# Blockage of the IL-31 Pathway as a Potential Target Therapy for Atopic Dermatitis

**DOI:** 10.3390/pharmaceutics15020577

**Published:** 2023-02-08

**Authors:** Raquel Leao Orfali, Valeria Aoki

**Affiliations:** Laboratory of Dermatology and Immunodeficiencies (LIM-56), Department of Dermatology, Faculdade de Medicina FMUSP, Universidade de Sao Paulo, Sao Paulo 01246-903, SP, Brazil

**Keywords:** interleukin-31 (IL-31), interleukin-31 receptor α-chain (IL-31RA), atopic dermatitis, pruritus

## Abstract

Atopic dermatitis (AD), a pruritic, inflammatory chronic disease with multifactorial pathogenesis, has been a therapeutic challenge. Novel target treatments aim to reduce not only the immunologic dysfunction and microbiome dysbiosis but also the recovery of the damaged skin barrier. The current review focuses on the interleukin 31 (IL-31) pathway and AD and offers an overview of the current clinical studies with monoclonal antibodies blocking this cascade. Pruritus, the key symptom of AD, has substantial participation of the IL-31 complex and activation of relevant signaling pathways. Epidermal keratinocytes, inflammatory cells, and cutaneous peripheral nerves express the interleukin-31 receptor α-chain (IL-31RA), upregulated by *Staphylococcus aureus* toxins or Th2 cytokines involved in AD. Nemolizumab is a humanized monoclonal antibody that antagonizes IL-31RA, inhibiting the IL-31 cascade and therefore contributing to reducing the pruritus and inflammation and recovering the damaged skin barrier in AD patients. Phases 2 and 3 clinical trials with nemolizumab in AD show a suitable safety profile, with a fast, efficient, and sustained reduction of pruritus and severity scores, especially when associated with topical treatment. Deciphering the full interplay of the IL-31 pathway and AD may expand the potential of nemolizumab as a targeted therapy for AD and other pruritic conditions.

## 1. Introduction

Atopic dermatitis (AD) is a chronic and pruriginous inflammatory skin disease with a high prevalence, especially in the pediatric population, affecting about 20% of children and 5–10% of adults. The pathogenesis of AD is multifactorial, with a complex interplay between immunological abnormalities, skin barrier defects, microbiome dysbiosis, and environmental triggers [1].

From a genetic point of view, patients may present with a predetermined defect of the skin barrier (e.g., filaggrin-FLG deficiency due to mutation of the *FLG* gene), which clinically manifests as dry skin (xerosis) [2]. Although this genetic defect of the skin barrier occurs in a fair number of patients, mutations of the *FLG* gene are not a key condition for the onset of AD since mutations are detected in about 20% of patients with moderate-to-severe AD, and about 50% of individuals with mutations in the *FLG* gene do not develop AD [3]. Furthermore, such skin barrier mutations may vary according to ethnic variations, being much more common among AD patients of European than Asian and African origins [4]. The skin barrier defects in AD involve not only decreased protein expression but also lipid alterations, including fatty acid elongase 3 and diacylglycerol o-acyltransferase 2 [5].

As a consequence of the deficient skin barrier, there is an increase in the expression of pro-inflammatory cytokines and, subsequently, activation of subsets of inborn lymphocytes and antigen-presenting cells (Th2 and Th22). IL-4 and IL-13 cytokines (Th2 cytokines) incite the recruitment of eosinophils and mast cells and secretion of IL-31 (itching-related cytokine). Environmental allergens and bacterial colonization (mainly *Staphylococcus aureus—S. aureus*) contribute to rupturing the skin barrier and perpetuating the skin inflammation seen in AD [6,7,8,9].

The set of phenotypic variations found in AD, such as age of onset, severity of the disease, association with atopic comorbidities (food allergies, rhinitis, asthma), and variable response to treatment, contributes to the characterization of specific endotypes [6]. The identification of possible environmental factors (e.g., pollution, UV radiation, humidity, among others) that may influence the course of the disease, also known as exosome, is extremely relevant for characterizing the development and progression of AD [6,10].

AD comprises one of the most prevalent diseases both in the pediatric and adult populations, with a high impact on the quality of life of patients and caretakers [11]. Those patients with moderate or severe disease often develop depression and anxiety and face difficulties in their daily scholarly or professional life [12]. The major symptom of AD is chronic itch, and its occurrence is related to sleep disturbances, reduced quality of life, and maintenance of the itch–scratch cycle [11]. In a recent study by Silverberg et al. [12], the severity of AD was related to sleep disturbances (SD), with reports from 40.7% of adults diagnosed with AD declaring at least one night of SD in the past week and 79.7% identifying at least some trouble sleeping in the past 3 days.

In the late adolescent population with AD, there are reports demonstrating that suicidal ideation was present in 15.5%, a significantly higher percentage than in non-AD patients; moreover, suicidal ideation in those with both AD and itch was 23.8% and was significantly associated [13], reinforcing the need to approach itch as a therapeutical target in AD. Stratification of neonates, children, and adult patients with AD in biomarker-based endotypes to distinguish the different forms of imbalance of the skin barrier, immune system, or microbial dysbiosis may contribute to the development of personalized therapeutic strategies [14]. A combination of accurate diagnosis with early targeted therapies is indispensable in addressing the heterogeneity of the disease [1]. 

The mechanisms of chronic pruritus in AD involve not only the defective skin barrier, with increased transepidermal water loss, but also dysregulation of the cutaneous immune response, generating a Th2-driven inflammatory response and sensitization of the cutaneous central and peripheral neurologic pathways [15,16,17]. Among the type 2 driven cytokines in AD, interleukin 31 (IL-31) and its receptor complex IL-31RA and oncostatin M receptor beta (OSMRβ) have been related to chronic itch [18]. 

Before initiating treatment-targeted strategies, our first step should include a detailed evaluation of the severity and extension of AD in order to establish a baseline disease burden [19]. Thus, early treatment strategies focused on the restoration of the skin barrier and microbiome and/or the targeting of Th2 inflammation, depending on the (endo)phenotype, is not only crucial for disease control but can also contribute to avoiding possible associated comorbidities and improvement in the quality of life of AD patients and their caretakers [20,21]. All these approaches together lead us to future opportunities for the patient to achieve an acceptable social, educational, and professional routine. Adequate adherence to treatment, with a potential impact on the reduction of cutaneous inflammation, will definitely contribute to the prevention or modification of the course of AD, as well as the reduction of associated comorbidities [22,23].

In this review, we will focus on the blockage of the IL-31 axis, describing its relevance in the control of the itch–scratch–inflammation cycle. The mechanism of action of monoclonal antibodies anti-IL31Rα involved in AD pathogenesis and its efficacy in reducing unstoppable pruritus, the major symptom of AD, will be discussed.

## 2. State of Art: IL-31 and AD Pathogenesis

AD pathogenesis comprises a complex interaction between a defective skin barrier, environmental factors, and a dysfunctional Th2 immune response, leading to a constant cutaneous inflammation state [15,24]. Numerous cells participate in this interface, including keratinocytes, dendritic cells, mast cells, basophils, eosinophils, macrophages, type 2 innate lymphoid cells (ILC2s), decreased epidermal barrier proteins (filaggrin, involucrin, claudin), antimicrobial peptides, T and B cells with its related cytokines (Th2 axis: IL-4, IL-13, IL-5, IL-10, TSLP, IL-31; increased Th17/IL-23 and Th22 axis; increased IgE levels), and an altered microbial diversity and predominance of *S. aureus* strains [15,25,26]. The Th2 axis has a particular role in AD pathogenesis, releasing multiple inflammatory and pro-inflammatory cytokines and initiating various immunoregulatory pathways, leading to an itch–scratch–inflammation circle with neuroimmune implications [25,27,28]. Although many aspects of the multifactorial immunopathogenesis of AD have been constantly analyzed, the scenario of heterogeneity in diverse populations still requires further investigation, reinforcing the immunological complexity of the disease and emphasizing gaps to be filled in terms of individualized treatment strategies [24].

IL-31 is one of the main cytokines linked to pruritus [29], the most prominent symptom of AD [15]. IL-31 is also an inducer of epidermal cell proliferation and skin remodeling and thickening in the chronic Th1-mediated phase [30,31]. IL-31, a cytokine of the IL-6 family, is secreted by many cell types, such as Th2 cells, monocytes, macrophages, keratinocytes, fibroblasts, dendritic cells, eosinophils, basophils, and cutaneous peripheral nerves [32]. IL-31 binds to its receptor IL-31RA and to OSMRβ, driving the activation of an inflammatory cascade via phosphorylation of JAK/STAT (Janus-activated kinase/signal transducer and activator of transcription) and PI3K/AKT (phosphatidylinositol 3-kinase/protein kinase) pathway. It is also capable of inducing the activation of the MAPK-JNK/p38 (mitogen-activated protein kinase-Janus kinase/p38) activation pathway [30]. However, the full comprehension of these mechanisms needs a better understanding.

As for the AD skin barrier, studies involving IL-31 revealed that it perpetuates the itch–scratch cycle in AD patients not only by upregulating inflammatory cells but also by direct modulation of the keratinocyte function. It is also capable of reducing the expression of skin barrier molecules such as filaggrin (FLG) and claudin-1 (CLDN1), impairing skin barrier function and augmenting exposure to antigens. As mutations of *FLG* have previously been associated with AD, the inhibitory effects of IL-31 on the expression of this skin barrier protein may be a key factor for AD individuals [28].

Regarding AD and neuroimmune communication, the IL-31 axis exerts a strong link between IL31RA expression, human DRGs (sensory nerves), skin-infiltrating mononuclear cells, and CD11b+ cells, corroborating its relevance in human AD [27,33]. A possible explanation for the feedback loop of AD inflammation indicates that IL-4, a Th2 cytokine, is involved in IL-31RA expression, increasing the production of the chemokines CCL17 and CCL22 by bone marrow-derived dendritic cells (BMDCs) and, together with external *stimuli* of the host’s defense, drives to a Th2 axis [30]. In synergy with IL-33, Il-31 is also capable of IL-6 induction, as well as other pro-inflammatory cytokines (CXCL1, CXCL10, CCL2, and CCL5). IL-31 is additionally associated with the elicitation of the surface production of the intercellular adhesion of molecule-1 (ICAM-1) on eosinophils and fibroblasts [30]. Staphylococcal exotoxins are also related to stimulating increased levels of IL-31RA expressed in human monocytes and macrophages [32,34]. Another mechanism of inflammation is related to staphylococcal exotoxins’ exposure: IL-31 upregulates pro-inflammatory cytokines in human macrophages and keratinocytes, and also mRNA encoding for antimicrobial peptides (AMPs) human β-defensin 2 and 3 [30,32,35]. Figure 1 illustrates the IL-31 pathway in AD inflammation.

Although the role of the IL-31/IL31RA axis in pruritus and AD is tightly established, the precise mechanisms of the role of IL-31 signaling in the pathogenesis of AD remain to be better clarified [27]. 

## 3. Blocking the IL-31 Pathway in AD

Considering the connection between IL-31 in the pathogenesis of AD and pruritus, a therapeutic approach targeting the IL-31 axis may expand the scope of AD treatment [36]. IL-31RA is expressed by immune cells such as activated macrophages, dendritic cells, eosinophils, basophils, epidermal keratinocytes, and cutaneous peripheral nerves, making it a specific target for the humanized monoclonal antibody nemolizumab. Blocking IL-31RA may reduce the IL-31 cascade, therefore modulating the inflammation and pruritus in AD. 

Blocking the IL-31 pathway and the itch involves a wide network of immune and pruritic mechanisms, including direct inhibition of the binding of IL-31 to cutaneous sensory neurons or downregulating the inflammatory Th2 and Th17 responses. There is also a modulation of keratinocyte proliferation and epidermal differentiation [37]. Moreover, transcriptomic changes seen in prurigo nodularis (PN) skin correlated with improvements in skin lesions and pruritus, with stabilization of extracellular matrix remodeling and processes associated with cutaneous nerve function [38]. 

In initial studies performed in BALB/c mice, IL-31 was injected in the murine ear to elicit scratching. Later, the animals received anti-mouse IL-31 receptor α-neutralizing antibody as treatment. The study demonstrated the efficacy of anti-IL-31 receptor α in decreasing ear thickening and dermatitis scores, showing its potential target as a treatment for itching related to AD [39]. In a subsequent study, cynomolgus monkeys demonstrated that a single subcutaneous injection of 1 mg/kg nemolizumab neutralizes IL-31 signaling and shows a suitable pharmacokinetic profile, suppressing the IL-31-induced scratching for about 2 months [40]. 

The first report of the nemolizumab trial in AD patients is from Japan. It shows that a single subcutaneous (SC) administration was well tolerated both in healthy volunteers and patients, decreasing pruritus, sleep disturbance, and topical use of hydrocortisone [41,42]. A phase 2 trial with SC nemolizumab in monthly doses showed significant improvement in pruritus in patients with moderate-to-severe AD [43,44]. All doses did not display superior responses to eczema area and severity index 75 (EASI-75) or 50 (EASI-50) when compared with placebo, even though the 0.5 mg/kg dose showed superior SCORAD-50 and SCORAD-75 responses [43,45].

Subsequently, there was a publication on a phase 2 part B trial, including a long-term extension study in adults with AD. One-hundred and ninety-one patients previously enrolled in phase 2 initial studies and joined phase 2 part B. The treatment regimen was based on the previous dose regimen of nemolizumab (0.1, 0.5, or 2.0 mg/kg Q4W or 2.0 mg/kg Q8W). The two major endpoints of this phase 2-B part were the percentage of improvement in pruritus from baseline using pruritus scales (visual analog scale or VAS) and improvement of AD severity scores based on EASI. The results showed suitable efficacy and a well-tolerated response of AD patients with refractory, moderate-to-severe AD and non-responsive to topical treatments after 64 weeks of follow-up [46,47].

A study on the impact of nemolizumab on work productivity and activity impairment in adults with moderate-to-severe AD inadequately controlled by topical treatments in a two-part, phase II, randomized control trial revealed that nemolizumab-treated patients reported improvements in their activities through week 64 [48]. The limitation of the study was the absence of an extension of the placebo arm up to week 64, therefore not allowing a comparison between the groups. Reported adverse effects included AD exacerbation, nasopharyngitis, peripheral edema, and elevated creatine kinase [46]. Currently, there are several ongoing phase I-III studies), (JapicCTI-173740, JapicCTI-173741, JapicCTI-183894, NCT03100344) in the pediatric population (patients aged 6 years and older) [49].

Another study performed with patients with moderate-to-severe AD that complained of severe, refractory pruritus utilized placebo versus subcutaneous nemolizumab (loading dose, followed by monthly 10, 30, or 90 mg until week 20, and a 12-week follow-up period until week 32), all arms associated with topical corticosteroids. The outcomes showed a reduction of EASI scores, immunoglobulin E (IgE) serum levels, and NRS-itch scores, with the best dose at 30 mg. Considering adverse events, there was a low incidence of peripheral edema and no serious adverse events. Two subjects on nemolizumab had early interruption due to increased creatine kinase levels [50,51].

Interestingly, one phase 2B study in moderate-to-severe AD patients as a post hoc analysis of patients with EASI scores ≥ 16 (moderate-to-severe AD), with endpoints that included change in EASI score at week 16, peak pruritus numeric rating scale (PPNRS), Investigator’s Global Assessment (IGA), changes in sleep and responders with ≥4-point improvement on PPNRS, showed that 68% of patients on nemolizumab achieved itch relief at day 2, sustained until week 16 with a significant response, when compared to those individuals on placebo. There was also improvement in sleep disturbances with nemolizumab 30 mg, with significant separation from placebo by day 3, and improvement with IGA. The drug proved to be safe, with nasopharyngitis and upper respiratory tract infection as the main side effects (Table 1) [52,53].

In Japan, a randomized, phase 3 clinical trial included AD patients with insufficient response to topical treatment and with pruritus (median VAS score of 75) and EASI score from 22.7 to 24.2. Patients received either nemolizumab (60 mg) or placebo every 4 weeks until week 16, with concomitant topical agents. Reduction of the VAS score was significant, exhibiting −42.8% in the nemolizumab group versus −21.4% in the placebo group (*p* < 0.001) at week 16. Secondary endpoints revealed a reduction of EASI score (−45.9% for the nemolizumab group versus −33.2% for the placebo group), and changes in daily VAS scores at day 15 (seen as early as day 2) were −10.3% for the nemolizumab group versus −4.4% for the placebo group [54,55]. Occurrence of injection-site reactions were further described in the nemolizumab group and compared to those who received a placebo (8% versus 3%, respectively). This trial demonstrated a greater reduction in pruritus than placebo over a period of 16 weeks in patients with AD with a poor response to topical agents and antihistamines. Limitations of the study included the short duration of treatment, the inclusion of a homogeneous population (100% Japanese patients), and the enrollment of patients with an age ≥ 13 years [54,55].

In the long term (≥52 weeks), phase III, multicentric studies with nemolizumab in AD patients with pruritus, not responsive to topical or oral treatments, the results confirmed that the combination of anti-IL-31Ra plus topical agents could improve or maintain the drug efficacy, with continuous improvement after week 16. Acute pruritus and AD flares were seldom reported during the 8-week follow-up period. The long-term treatment of nemolizumab in association with topical agents is highlighted in this study, strengthened by evidence that combination therapies enhance better outcomes in patients with AD and inadequate control of moderate-to-severe pruritus [56].

Of note, the AD adolescent population with moderate-to-severe disease has already been enrolled in open-label studies with nemolizumab (ages 12–17). One open-label study in adolescents included patients with baseline EASI > 16, IGA > 3 and BSA > 10% and daily peak pruritus numeric rating scale (PPNRS) intensity >4, who received subcutaneous nemolizumab (loading dose 60 mg and 30 mg every 4 weeks until week 12, associated with topical treatment (calcineurin inhibitors or corticosteroids). There was a visible improvement of the rash, itch, and sleep, with a reduction of EASI, PPNRS, and rating scale similar to the results achieved in AD in adults with the same dose regimen [57].

Pioneer approval of nemolizumab for treating itch associated with atopic dermatitis for adolescents and adults occurred in Japan (28 March 2022) [37]. However, the indication is recommended only for those patients that are poor responders to existing treatments. Moreover, the outcomes of clinical trials with nemolizumab emphasize the beneficial effect of concomitant use of adjuvant topical treatments such as corticosteroids, calcineurin inhibitors, or even emollients, therefore improving the skin inflammation and the itch cycle, as well as the patient’s quality of life [58,59]. Figure 2 depicts the timeline of nemolizumab. 

Table 2 illustrates the panel of ongoing or completed clinical trials with nemolizumab in the USA, Europe, and Japan. To date, there are 10 completed, 10 ongoing, and 3 recruiting studies.

## 4. Discussion

The expansion of targeted therapies for atopic dermatitis opens a broad and novel perspective for the treatment of atopic dermatitis. The complex pathophysiology of AD demands a combination of treatment modalities that vary from the identification of triggering factors to conventional or novel systemic therapies in order to recover the skin barrier defects, microbiome dysbiosis, and immune dysfunction in AD, minimizing the most important symptom of AD: the pruritus [11].

There is an imperative need to further investigate the mechanisms of inflammation and itch and to evaluate the efficacy and safety of new monoclonal antibody or small molecule therapy in AD. AD may start as a skin condition, but it may evolve into a Th2 systemic disease, with associated atopic and non-atopic comorbidities and a heterogeneous pattern of the immune response, depending on the skin barrier defects, age, and ethnicity [3,11,15,16,63,64,65,66,67,68].

Results from clinical trials with SC anti-IL-31RA (nemolizumab) in AD patients demonstrate that it suppresses the pruritus in a very fast manner (beginning at day 2 with continued improvement to week 16), reduces the severity scores, and its effectiveness improves over time, usually achieving better outcomes after week 16 in the 30 mg dose every 4 weeks [53]. In a systematic review and meta-regression analysis of randomized clinical trials, the results showed that nemolizumab significantly decreased the pruritus VAS score (WMD = −18.86, 95% CI: −27.57 to −10.15, *p* < 0.001; I^2^ = 56.2%, *p* heterogeneity = 0.005), and EASI scores (WMD = −11.76, 95% CI: −20.55 to −2.96, *p* = 0.009; I^2^ = 0%, *p* heterogeneity = 0.978), when compared to the placebo group [59]. The use of concomitant topical corticosteroids, calcineurin inhibitors, or emollients and no severe adverse events offered better results [28,46,54,59]. In studies with other immunobiological agents such as dupilumab, an anti-IL-4 receptor a monoclonal antibody, there is an improvement of EASI score (−68% in dupilumab 300 mg every 2 weeks vs. −18.0% in the placebo group) [69]. A network meta-analysis by Silverberg et al. that included 19 phase 2 and phase 3 randomized controlled trials with abrocitinib, baricitinib, dupilumab, lebrikizumab, nemolizumab, tralokinumab, and upadacitinib showed that in monotherapy RCTs, upadacitinib 30 mg once daily (QD) achieved the best efficacy (83.6% achieved EASI-50 response), while, in combination therapy RCTs, the highest EASI-50 was achieved by abrocitinib 200 mg QD (86.6%) [70]. Nemolizumab may not improve eczematous lesions as much as the other immunobiologics or JAK inhibitors, but blockage of IL-31 improves pruritus more efficiently and rapidly. Thus, the use of nemolizumab in itch–scratch cycle-associated lichenoid skin lesions may be a suitable therapeutic target by blocking the IL-31 axis [32].

Even though the blockage of IL-31, a Th2 interleukin involved in AD, would induce improvement of inflammation and pruritus, there are still unclear points to be addressed when blocking the IL-31 pathway; IL-31 antagonism in AD treatment needs more elucidation once it is not clear whether its effects include only symptomatic, antipruritic relief [71]. There are reports of the use of nemolizumab for AD and adverse events such as AD and asthma exacerbation, apart from upper respiratory tract infection, nasopharyngitis, peripheral edema, elevated creatine kinase, and a reaction at the injection site [28]. Therefore, additional investigation on the consequences of the blockage of the IL-31 pathway is a relevant point to be further explored. 

## 5. Conclusions

It is expected that monoclonal antibodies targeted against cytokines deeply involved in the pathogenesis of AD, such as Th2 cytokines (IL-4, IL13, and IL-31), have the potential to be used as novel treatments for AD. Nemolizumab is an anti-IL-31 RA humanized monoclonal antibody that rapidly reduces itch in atopic dermatitis patients, with sustained efficacy, shows a suitable safety profile and decreases the itch–scratch cycle associated with AD patients, improving their quality of life. A better understanding of the involved mechanisms of the IL-31 axis in AD and additional long-term, real-world, and head-to-head comparative studies are necessary to expand its use in other pruritic conditions, especially in combination with other therapeutic modalities. 

## 6. Future Directions

Since AD is a clinically heterogeneous disease, it is of extreme relevance to better identify the subtypes of AD, which will profit the most from the current and future specific therapies, including monoclonal antibodies directed to T cell cytokines such as IL-4, IL-13, and IL-31, for the best individualized benefit. Targeting the elements of the IL-31 pathway may be beneficial for treating intractable itch in AD and other pruritic diseases, such as prurigo nodularis, either as a mono or combined therapy.

## Figures and Tables

**Figure 1 pharmaceutics-15-00577-f001:**
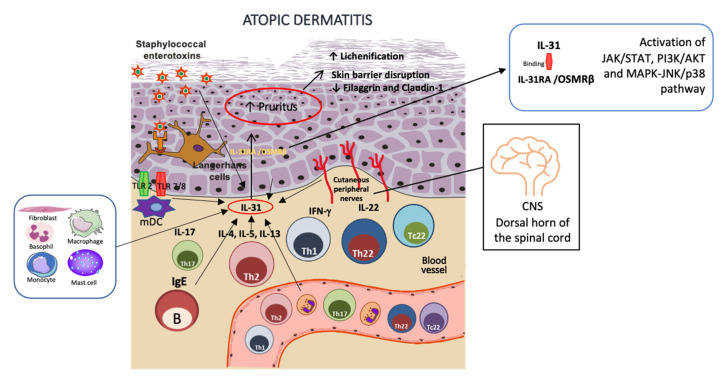
IL-31 role in AD skin inflammation. IL-31 is secreted by many cell types, such as Th2 cells, B cells, monocytes, macrophages, keratinocytes, fibroblasts, dendritic cells, eosinophils, basophils, and cutaneous peripheral nerves, with neuroimmune implications. It induces epidermal cell proliferation and thickening in the chronic Th1-mediated phase, increasing lichenification. Exposure to Staphylococcal exotoxins increases IL-31 release in macrophages and keratinocytes. After binding to its receptor, IL-31RA, and to OSMRβ heterodimer, it initiates an inflammatory cascade via phosphorylation of JAK/STAT, PI3K/AKT, and MAPK-JNK/p38 activation pathway, indicating a circle of itch–scratch–inflammation in AD skin. The figure was partly generated using Servier Medical Art, provided by Servier, licensed under a Creative Commons Attribution 3.0 unported license.

**Figure 2 pharmaceutics-15-00577-f002:**
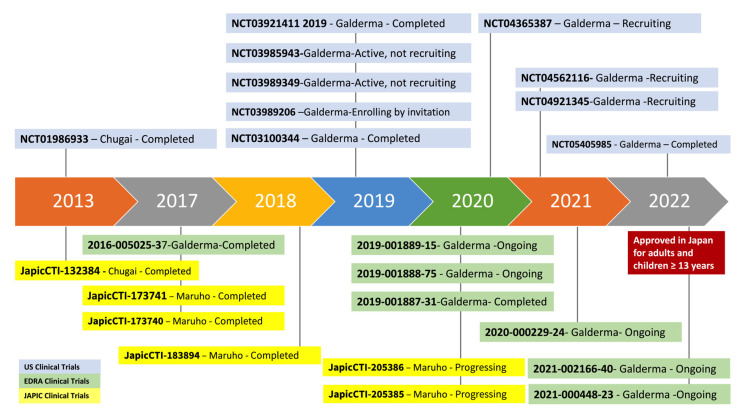
Timeline of clinical trials with Nemolizumab. Graphic timeline illustration of clinical trials involving nemolizumab both in adult and pediatric populations, described as: clinical trial number ID, sponsor, and current situation of the study, updated in 5 January 2023 according to https://clinicaltrials.gov (accessed on 5 January 2023) [60], https://www.clinicaltrialsregister.eu/ctr-search/search/ (accessed on 5 January 2023) [61], and https://rctportal.niph.go.jp (accessed on 5 January 2023) [62] websites. U.S.: United States; EudraCT: European Union Drug Regulating Authorities Clinical Trials Database; JAPIC: Japan Pharmaceutical Information Center.

**Table 1 pharmaceutics-15-00577-t001:** Nemolizumab most related side effects.

Study	Main Side Effects Associated	Main Outcomes
Nemoto, O, et al. (2016) [41]—Part A	No deaths, serious side effects, or discontinuation Infectious enteritis ↑ C-reactive protein Nasopharyngitis	- Well tolerated with satisfactory safety profile - Most reported serious adverse event was aggravation of AD - Concerns about nemolizumab in patients with pre-existing severe asthma - Minimal side effects improved as treatment continued - Suitable safety and tolerability of nemolizumab for AD associated with incontrollable pruritus [28]
Nemoto, O, et al. (2016) [41]—Part B	↑ blood CK ↑ ALT/AST Pharyngitis Myalgia	
Nemoto, O, et al. (2016) [41]—Part C	Exacerbation of AD Folliculitis Nasopharyngitis	
Ruzicka, T et al. (2017) [43]	Exacerbation of AD Nasopharyngitis Upper respiratory tract infection Peripheral edema ↑ blood CK	
Silverberg et al. (2020) [50]	AD exacerbation ↑ incidence of asthma (in subjects with previous history of asthma) ↑ CK levels	
Kabashima et al. (2018) [46]	Nasopharyngitis Exacerbation of AD ↑ blood CPK Upper respiratory tract infection Headache Peripheral edema Impetigo Nasopharyngitis Injection-site reaction	
Kabashima et al. (2020) [54]	Worsening of atopic dermatitis Meniere’s disease Alopecia Peripheral edema Injection-site reaction ↑ TARC Musculoskeletal and connective tissue symptoms	

AD: atopic dermatitis; CK: creatine kinase; ALT: alanine transaminase; AST: aspartate aminotransferase; CPK: creatine phosphokinase; TARC: thymus and activation-regulated chemokine.

**Table 2 pharmaceutics-15-00577-t002:** Nemolizumabs’clinical trials status in USA, Europe, and Japan.

Clinical Trial ID	Study	Start Date	Status	Region
NCT01986933	A Phase 2 Study of CIM331 for Atopic Dermatitis Patients	2013	Completed	USA *
NCT03100344	Dose-ranging Study of Nemolizumab in Atopic Dermatitis	2019	Completed	
NCT03989206	Long-term safety and efficacy of Nemolizumab with moderate-to severe atopic dermatitis	2019	Enrolling by invitation	
NCT03989349	Efficacy & Safety of Nemolizumab in Subjects with Moderate-to-Severe Atopic Dermatitis	2019	Active, not recruiting	
NCT03985943	Efficacy and Safety of Nemolizumab in Subjects with Moderate-to-Severe Atopic Dermatitis	2019	Active, not recruiting	
NCT03921411	A Pharmacokinetics and Safety Study of Nemolizumab in Adolescent Participants with Atopic Dermatitis (AD)	2019	Completed	
NCT04365387	A Study to Assess Immunization Responses in Adult and Adolescent Participants with Moderate-to-Severe Atopic Dermatitis Treated with Nemolizumab	2020	Recruiting	
NCT04921345	Pharmacokinetics, Safety and Efficacy of Nemolizumab in Participants with Moderate-to-Severe Atopic Dermatitis-	2021	Recruiting	
NCT04562116	A Study to Assess the Effects of Nemolizumab on Cytochrome P450 Substrates in Participants with Moderate-to-Severe Atopic Dermatitis	2021	Recruiting	
NCT05405985	Study to Access the Relative Bioavailability of Subcutaneous Dose of Nemolizumab When Administered Via Auto-Injector Versus Dual-Chamber Syringe	2022	Completed	
2016-005025-37	A randomized, double-blind, multi-center, parallel-group, placebo-controlled dose-ranging study to assess the efficacy and safety of nemolizumab (CD14152) in moderate-to-severe atopic dermatitis subjects with severe pruritus receiving topical corticosteroids	2017	Completed	Europe (EudraCT) **
2019-001889-15	A Prospective, Multicenter, Long-Term Study to Assess the Safety and Efficacy of Nemolizumab (CD14152) in Subjects with Moderate-to-Severe Atopic Dermatitis	2020	Ongoing	
2019-001888-75	A Randomized, Double-Blind, Placebo-Controlled Study to Assess the Efficacy and Safety of Nemolizumab (CD14152) in Subjects with Moderate-to-Severe Atopic Dermatitis	2020	Ongoing	
2019-001887-31	A Randomized, Double-Blind, Placebo-Controlled Study to Assess the Efficacy and Safety of Nemolizumab (CD14152) in Subjects with Moderate-to-Severe Atopic Dermatitis	2020	Completed	
2021-002166-40	A Randomized, Double-Blind, Placebo-Controlled Study to Assess the Efficacy and Safety of Nemolizumab in Subjects with Moderate-to-Severe Atopic Dermatitis with Inadequate Response to or for Whom Cyclosporine A is not Medically Advisable	2022	Ongoing	
2020-000229-24	An Open-label Drug-Drug Interaction Study to Assess the Effects of Nemolizumab on Cytochrome P450 Substrates in Subjects with Moderate-to-Severe Atopic Dermatitis	2021	Ongoing	
2021-000448-23	A Multicenter, Open-Label, Single-Group Clinical Trial to Assess the Pharmacokinetics, Safety and Efficacy of Nemolizumab (CD14152) in Pediatric Subjects (aged 2 to 11 years) with Moderate-to-Severe Atopic Dermatitis	2022	Ongoing	
JapicCTI-132384	A Phase 2 Study of CIM331 for Atopic Dermatitis Patients	2013	Completed	Japan (JAPIC) ***
JapicCTI-173741	A Phase I, Open-label Single Dose Escalating Study to Evaluate the Safety, Tolerability, and Pharmacokinetics of Nemolizumab in Pediatric Patients with Atopic Dermatitis	2017	Completed	
JapicCTI-173740	A Phase III, Randomized, Double-blind, Placebo-controlled, Multi-Center Study to Evaluate the Efficacy and Safety of Nemolizumab in Japanese Atopic Dermatitis Patients with moderate to severe pruritus	2017	Completed	
JapicCTI-183894	A Phase III, Open-label, Multi-Center Study to Evaluate the long-term Safety of Nemolizumab in Japanese Atopic Dermatitis Patients with moderate to severe pruritus	2018	Completed	
JapicCTI-205386	A Phase I, Open label, Multi-Center Clinical Pharmacological Study of Nemolizumab with a Single-dose in Japanese Atopic Dermatitis Patients with moderate to severe pruritus	2020	Progressing	
JapicCTI-205385	A Phase III, Randomized, Double-blind, Placebo-controlled, Multi-Center Study to Evaluate the Efficacy and Safety of Nemolizumab in Japanese Pediatric Atopic Dermatitis Patients with moderate to severe pruritus	2020	Progressing	

EudraCT: European Union Drug Regulating Authorities Clinical Trials Database; JAPIC: Japan Pharmaceutical Information Center. * Consulted on in https://clinicaltrials.gov website (accessed on 5 January 2023) [60]; ** Consulted on https://www.clinicaltrialsregister.eu/ctr-search/search/ (accessed on 5 January 2023) website [61]; *** Consulted on https://rctportal.niph.go.jp (accessed on 5 January 2023) website [62].

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
