# Peer review of "Blockage of the IL-31 Pathway as a Potential Target Therapy for Atopic Dermatitis"

_pharmaceutics, 2023, doi:10.3390/pharmaceutics15020577_

Round 1

Reviewer 1 Report

In this review, Orfali et al summarized interleukin 31 (IL-31) pathway and AD and offered an overview of the current clinical studies with monoclonal antibodies blocking this cascade. Deciphering the full interplay of IL-31 pathway and AD may expand the potential of nemolizumab as a target therapy for AD and other pruritic conditions. The review is so clearly organized with easy-to-understand diagrams about the role of IL-31 in skin immunity that the contents are so easy to understand that one can get the contents in one's head just by reading it. The review also summarizes the clinical trials that have been conducted to date in various countries. It is very easy to understand what kind of clinical trials have been conducted up to now and what kind of clinical trials are being conducted now. From my reading, I did not find any particular flaws in this paper. I felt that the content was presented from a wide range of perspectives.

If I may make a suggestion, in Table 2, since we have the results of clinical trials that have been conducted to date, I would like to point out that there are common indicators in each trial, such as the EASI-50 and the itchiness score. You could integrate the results of all the clinical trials (of course, it may be difficult to summarize the results because they differ depending on whether topical agents are used or not, open/double-blinded, etc.) and present a meta-analysis of nemolizumab's effect on skin rash and itching.
It would be better if a meta-analysis of the effect of nemolizumab on skin rash and itching could also be presented.

minor concerns)

1) In line 120, "PI3K/AKT (phosphatidylinositol 30-kinase/protein kinase) pathway," should it be "3" instead of "30"? Please correct as appropriate.

2) In line 163, the sentence "IL-31RA, expressed in immune cells, keratinocytes, macrophages, cutaneous is a specific target for nemolizumab, a humanized monoclonal antibody." I found it difficult to understand the meaning of the text. Please correct as appropriate.

3) In line 178, "nemolizumab neutralizes c IL-31 signaling", is the c unnecessary? Please correct as appropriate.

Author Response

Dear reviewers:

The authors would like to thank you for your precious time to evaluate this paper. Your expertise is very much appreciated, and we would like to thank you for the suggestions. Our point-by-point replies are below listed:

Review 1

Comments and Suggestions for Authors

In this review, Orfali et al summarized interleukin 31 (IL-31) pathway and AD and offered an overview of the current clinical studies with monoclonal antibodies blocking this cascade. Deciphering the full interplay of IL-31 pathway and AD may expand the potential of nemolizumab as a target therapy for AD and other pruritic conditions. The review is so clearly organized with easy-to-understand diagrams about the role of IL-31 in skin immunity that the contents are so easy to understand that one can get the contents in one's head just by reading it. The review also summarizes the clinical trials that have been conducted to date in various countries. It is very easy to understand what kind of clinical trials have been conducted up to now and what kind of clinical trials are being conducted now. From my reading, I did not find any particular flaws in this paper. I felt that the content was presented from a wide range of perspectives.

R: Thank you for your kind comments.

If I may make a suggestion, in Table 2, since we have the results of clinical trials that have been conducted to date, I would like to point out that there are common indicators in each trial, such as the EASI-50 and the itchiness score. You could integrate the results of all the clinical trials (of course, it may be difficult to summarize the results because they differ depending on whether topical agents are used or not, open/double-blinded, etc.) and present a meta-analysis of nemolizumab's effect on skin rash and itching. 
It would be better if a meta-analysis of the effect of nemolizumab on skin rash and itching could also be presented.

R: Thank you very much for your suggestion. We thought about including this point, but by reviewing the literature, we detected an excellent meta-analysis with these indicators, recently published in 2022. Therefore, we added a specific commentary in the discussion regarding  your suggestion, as follows:

“In a systematic review and meta-regression analysis of randomized clinical trials, the results showed that nemolizumab significantly decreased the pruritus VAS score (WMD = −18.86, 95% CI: −27.57 to −10.15, p < 0.001; I2 = 56.2%, pheterogeneity = 0.005), and EASI scores (WMD = −11.76, 95% CI: −20.55 to −2.96, p = 0.009; I2 = 0%, pheterogeneity = 0.978), when compared to the placebo group.”

Ref: Liang, J.; Hu, F.; Dan, M.; Sang, Y.; Abulikemu, K.; Wang, Q.; Hong, Y.; Kang, X. Safety and Efficacy of Nemolizumab for Atopic Dermatitis With Pruritus: A Systematic Review and Meta-Regression Analysis of Randomized Controlled Trials. Front Immunol 2022, 13, 825312, doi:10.3389/fimmu.2022.825312.

Minor concerns

1) In line 120, "PI3K/AKT (phosphatidylinositol 30-kinase/protein kinase) pathway," should it be "3" instead of "30"? Please correct as appropriate.

R:Thank you for your comment. Typo corrected in the main text.

2) In line 163, the sentence "IL-31RA, expressed in immune cells, keratinocytes, macrophages, cutaneous is a specific target for nemolizumab, a humanized monoclonal antibody." I found it difficult to understand the meaning of the text. Please correct as appropriate.

 R:Thank you for your comment. We rephrased the context:

 “IL-31RA  is expressed by immune cells such as activated macrophages, dendritic cells, eosinophils, basophils, epidermal keratinocytes and cutaneous peripheral nerves, making it a specific target for the humanized monoclonal antibody nemolizumab.”

3) In line 178, "nemolizumab neutralizes c IL-31 signaling", is the c unnecessary? Please correct as appropriate.

R: Thank you for your comment- this was a typo. The correct should be cynomolgus IL-31, not c IL-31. But how cynomolgus was already mentioned in the beginning of the sentence, we decided to exclude repetition. Edited in the main text.

Reviewer 2 Report

This is a well written paper on the efficacy of IL-31 pathway blockage and potential implications for further research.

The paper is presented in a clear and concise way.

I have no critical comments to give.

Author Response

Review 2

Comments and Suggestions for Authors

This is a well written paper on the efficacy of IL-31 pathway blockage and potential implications for further research.

The paper is presented in a clear and concise way.

I have no critical comments to give.

R:Thank you for your kind comments.

Reviewer 3 Report

The paper presents in detail the importance of interleukin-31 in the pathogenesis of AD, as well as the current data on attempts to use it as a therapeutic target. 

The manuscript is detailed and contains the latest data. 

Here are just a few suggestions for possible consideration: 

Page 2 line 49: I thnik that bacterial colonizstion would be more appropriate than infection.

The following points should be addressed in the discussion

- the speed of itch relief

- the impact of other biological drugs, new low molecular weight substances such as JAK inhibitors on the IL-31 pathway

- exacerbation of asthma as a side effect

Author Response

Review 3

Comments and Suggestions for Authors

The paper presents in detail the importance of interleukin-31 in the pathogenesis of AD, as well as the current data on attempts to use it as a therapeutic target. 

The manuscript is detailed and contains the latest data. 

Here are just a few suggestions for possible consideration: 

Page 2 line 49: I thnik that bacterial colonizstion would be more appropriate than infection.

R: Thank you for your suggestion. Edited in main text.

The following points should be addressed in the discussion

- the speed of itch relief

- the impact of other biological drugs, new low molecular weight substances such as JAK inhibitors on the IL-31 pathway

- exacerbation of asthma as a side effect

R:Thank you for your suggestions. Please see the replies to your questions/suggestions:

-Concerning the speed of itch relief, this subject is already discussed, but we added the info of specific time of itch relief in lines: 308-311.

-Concerning the impact of other biological drugs, the role of new low molecular weight JAK inhibitors on the IL-31 pathway is presented  in lines: 315-329.

-The exacerbation of asthma as a side effect is mentioned in line 335.
